# Succinate as a New Actor in Pluripotency and Early Development?

**DOI:** 10.3390/metabo12070651

**Published:** 2022-07-15

**Authors:** Damien Detraux, Patricia Renard

**Affiliations:** Laboratory of Biochemistry and Cell Biology (URBC), Namur Research Institute for Life Sciences (NARILIS), University of Namur (UNamur), 5000 Namur, Belgium; damien.detraux@unamur.be

**Keywords:** embryonic stem cells, naïve, primed, 2-cell like cells, succinate, SUCNR1

## Abstract

Pluripotent cells have been stabilized from pre- and post-implantation blastocysts, representing respectively naïve and primed stages of embryonic stem cells (ESCs) with distinct epigenetic, metabolic and transcriptomic features. Beside these two well characterized pluripotent stages, several intermediate states have been reported, as well as a small subpopulation of cells that have reacquired features of the 2C-embryo (2C-like cells) in naïve mouse ESC culture. Altogether, these represent a continuum of distinct pluripotency stages, characterized by metabolic transitions, for which we propose a new role for a long-known metabolite: succinate. Mostly seen as the metabolite of the TCA, succinate is also at the crossroad of several mitochondrial biochemical pathways. Its role also extends far beyond the mitochondrion, as it can be secreted, modify proteins by lysine succinylation and inhibit the activity of alpha-ketoglutarate-dependent dioxygenases, such as prolyl hydroxylase (PHDs) or histone and DNA demethylases. When released in the extracellular compartment, succinate can trigger several key transduction pathways after binding to SUCNR1, a G-Protein Coupled Receptor. In this review, we highlight the different intra- and extracellular roles that succinate might play in the fields of early pluripotency and embryo development.

## 1. Introduction

The successful isolation of cells from inner cell mass (ICM) of the mouse pre-implantation blastocyst by Evans, Kaufman and Martin [1,2] in 1981 enabled scientists to investigate in vitro the molecular mechanisms that are the root of pluripotency—the ability to form cells from all three germ-layers. This broad area of research aimed at a better characterization of the early developmental steps that could be further used to model various cell types and organ differentiation for cell therapies. The pluripotent cells emerging from this pre-implantation blastocyst represent a naïve pluripotent state with epigenetic, metabolic and transcriptomic features resembling the preimplantation blastocyst [3,4,5]. However, pluripotency has been considered as a continuum, including the primed pluripotency resembling the post-implantation epiblast [6,7], the formative state [8], in between or even alternative states such as the poised state [9] or the paused pluripotent state mimicking embryonic diapause [10,11]. The naïve mESC culture was also reported to present some heterogeneity, with some cells re-acquiring features of the 2C-embryo (2C-like cells, 2CLCs) such as the re-expression of retrotransposons or 2C-specific genes [12,13]. Interestingly, these cells are able to colonize the extraembryonic tissues when implanted in chimeras, demonstrating their “totipotent-like” capacity [12].

One of the most striking features distinguishing these pluripotent states is the drastic metabolic remodeling occurring during their transitions, reminiscent of the changes observed during development (extensively reviewed in [14,15,16]). For example, while naïve ESCs rely on a bivalent metabolism, using both the oxidative phosphorylations (OXPHOS) and glycolysis, primed ESCs are mostly glycolytic with little to no mitochondrial respiration observed [17,18]. This drop in OXPHOS activity is already observed within the formative state [8]. This metabolic difference can be the driver of the transition since the activation of OXPHOS enhances the reprogramming of primed murine induced pluripotent stem cells (iPSC) into the naïve state, while inhibition of the OXPHOS activity through the activity of Lin28 (lin-28 homolog A) or HIF1α (Hypoxia-inducible factor 1-α) pushes naïve cells forward [18,19,20]. As the metabolic hub of the cell, the mitochondrion is a key player in this transition and, interestingly, its morphology changes dramatically during this conversion. Albeit using their mitochondria and their electron transport chain (ETC) complexes at a higher rate, naïve cells do not possess mature mitochondria. It is only during the transition that this organelle goes from a round shape with sparse and irregular cristae to more elongated mitochondria with well-defined transverse cristae. All these changes were described as one of the major hallmarks of the implantation of the embryo in humans [18], mice [17] or even dogs [21]. This morphological and metabolic remodeling occurring during the naïve-to-primed transition is also observed during stem cell differentiation and commitment, during which the cell re-acquires an OXPHOS-based metabolism or conversely switches to a glycolytic based metabolism during reprogramming into iPSCs (reviewed in [22,23]).

Aside from a change in OXPHOS utilization, the beta oxidation of fatty acids is also strongly reduced during the naïve-to-primed transition of human and murine cells, partly due to CPT-1 (carnitine palmitoyltransferase 1)-dependent blockage of fatty acid import to the mitochondrial matrix, due to CPT1A downregulation by micro RNAs (miRNAs) and repressive chromatin marks, as shown with the oxygen consumption rate when the cells are presented with palmitate (C16:0) [18].

Besides its role in energy production, the mitochondrial function is also essential to provide the metabolites such as α-ketoglutarate (αKG) [24,25] or even S-adenosyl methionine (SAM) [18] that are necessary for epigenetics modifying enzymes (reviewed in [26]) also playing a key role during the transition between pluripotent states (reviewed in [27]). More recently, the activity of a non-canonical cytosolic tricarboxylic acid (TCA) cycle was also shown to be crucial for pluripotency maintenance and exit [28]. While the role of acetyl-CoA and αKG are probably the most thoroughly studied, other metabolites also need to be considered. Due to its many roles in cellular functions, this review focuses on succinate and how it can govern early development and cell fate control.

## 2. Succinate as Metabolic Cross-Roads

As a TCA cycle intermediate, succinate is one of the crucial metabolites of the cell (Figure 1). In this cycle, succinyl-CoA, along with CO_2_ and nicotinamide adenine dinucleotide (NADH), is the product of the oxoglutarate dehydrogenase complex (OGDC), also known as the alpha-ketoglutarate dehydrogenase complex (α-KGDH). Then, succinyl-CoA is taken up by the succinate-CoA ligase SUCL, composed of a heterodimer of an invariant α subunit (SUCLG1) and a β subunit, either SUCLA2 or SUCLG2. Depending on the association with SUCLA2 or SUCLG2, the products will be the substrate level phosphorylation of an ADP or a GDP along with the release of a succinate molecule. Finally, the uptake of succinate by the succinate dehydrogenase complex (SDH) is at the crossroads of the TCA and the ETC since the SDH is the complex II of the ETC. The reaction catalyzed by SDH will consume the succinate, forming fumarate and FADH_2_ (Flavin adenine dinucleotide). 

Further involved in the energy metabolism, succinate and succinyl-CoA are also the entry point in the TCA for anaplerotic reactions from branched-chain amino acids or even from propionate, reinforcing the role of this metabolite as a key node in mitochondrial metabolism. In addition, and in combination with glycine, succinyl-CoA is also the starting point to the heme biosynthesis through the formation of successive porphyrin intermediates. The process of heme biosynthesis has the particularity of consuming a large amount of succinate molecules, for two reasons. First, eight moles of succinyl CoA and glycine are stoichiometrically required to synthetize one mole of heme. Second, this theoretic ratio is actually largely underestimated as the heme synthesis pathway is somewhat inefficient. Indeed, precursors are not always converted into heme, and some excess porphyrins and side products are thus degraded or excreted [29]. It is estimated that 2nmol/day of porphyrins are excreted in rats [30] and this estimation is multiplied by 100 to 1000 for humans [31]. As a consequence, the heme biosynthesis is funneling down succinyl-CoA from the TCA. In the case of porphyria, the imbalance in heme synthesis observed was linked to a change in the succinyl-CoA availability [32].

Finally, it is worth mentioning that succinate and succinyl-CoA are also produced outside of mitochondria, namely in the peroxisome. Indeed, a peroxisome specific thioesterase (ACOT4) produces succinate out of the succinyl-CoA formed by the peroxisomal beta- or omega-oxidation of fatty acids [33], exiting the organelle possibly through the peroxisomal membrane protein 2 (PXMP2). Interestingly, it has been shown that this production of succinate by the peroxisome in response to an increase of free fatty acids (FFAs) could lead to an impairment of mitochondrial fatty acid β-oxidation through an increase of the NADH/NAD^+^ ratio [34]. This succinate-mediated mitochondrial β-oxidation impairment could have implications in the naïve-to-primed ESC transition as this pathway has been shown to be upregulated in naïve ESCs [18,35] and in the ICM of pre-implantation blastocysts [36].

## 3. Regulatory Roles of Intracellular Succinate

Succinate’s effects extend well beyond a role in mitochondrial ATP production, and include an action in protein post-translational modification, a regulation of the epigenetic landscape and an action as a paracrine signal to the neighboring cells. The last two effects outside the mitochondria are dependent on the release of the metabolite from the organelle (Figure 1). On the one hand, succinate exported through the inner mitochondrial membrane (IMM) and the outer mitochondrial membrane (OMM) is dependent on the mitochondrial dicarboxylate carrier (mDIC; *SLC25A10*) and the voltage-dependent anion-selective channel protein (VDAC1), respectively [37,38]. On the other hand, secretion in the extracellular space is mediated mainly by the monocarboxylate transporter MCT1 (*SLC16A1*), using the protonated form of succinate [39,40], while the organic anion transporters (OATs; *SLC22* family) are exchanging entering organic anions against succinate [41]. Finally, extracellular succinate can be imported to the cytosol via sodium-dependent carboxylate transporters of the *Slc13* family. So far, three members have been described with this function, the Na^+^/dicarboxylate cotransporter 1 (NaDC1; *SLC13A2*) and three (NaDC3; *SLC13A3*) with the highest affinity for succinate, and the Na^+^/citrate cotransporter (NaCT; *SLC13A5*). These transporters are mainly expressed in the intestine, liver or kidney [42,43] but their expression is also reported in RNAseq data of ESCs and embryos [8,18,44,45,46,47,48].

Originally produced as a metabolite from the TCA cycle in the mitochondria, succinate and its activated form (succinyl-CoA) can exert several regulatory effects all across the cellular compartments. Indeed, in the cytosol, succinate can inhibit the activity of α-ketoglutarate-dependent dioxygenases (2OGX) enzymes such as the PHD enzymes, regulating the stability of the hypoxia-inducible factor alpha (HIFα) subunits and thus their activity on the hypoxia-response elements (HRE). Succinate and its activated form could also induce the succinylation of lysine residues of various proteins, through the activity of succinylases such as the carnitine palmitoyltransferase I (CPT1) at the mitochondrial surface or the histone acetyltransferase 1 (HAT1) and the histone acetyltransferase 2A (KAT2A) in the nucleus. These succinyl-lysine marks are removed by the sirtuins 5 or 7 (Sirt5 or 7). In the nucleus, succinate inhibits histone demethylases (HDMs) and DNA demethylases of the Ten-eleven translocation (TET) family to regulate histone or DNA methylation levels, respectively. Finally, in the extracellular space, succinate can activate its receptor SUCNR1. 

### 3.1. Protein Lysine Residue Succinylation

While protein succinylation on lysine residues after in vitro incubation with succinic acid has been previously demonstrated, the natural occurrence of this post-translational modification was only demonstrated in vivo 10 years ago [49]. Chemically speaking, the impact of such a modification, compared for example with acetylation, is more drastic. Indeed, the two negative charges of the succinyl moiety switch the net charge of the positive lysine residue to a net negative one. Other than a charge modification, the succinylation of a lysine implies a relatively big structural change by adding a mass of 100 Daltons [49]. In the same way as protein acetylation, protein succinylation is a direct link between the mitochondrial metabolic activity (TCA and ETC) to protein functions, even in remote cellular compartments.

So far, the mechanisms leading to protein succinylation are not fully understood and could be enzymatic or non-enzymatic [50]. However, several enzymes exhibit a succinyltransferase capacity such as KAT2A [51], CPT-1A [52] or HAT1 [53]. Experimentally, hypersuccinylation can be provoked by mutating the SDH complex, resulting in the accumulation of succinate that is able to exit mitochondria [54], or by inhibiting the sirtuin enzymes responsible for protein desuccinylation. Sirt5, primarily located in the mitochondrial matrix, and Sirt7, located in the nucleus, belong to this category of enzymes [55,56]. Overall, and depending on the context such as Sirt5^−/−^ or SDH^−/−^ cells, up to 2000 succinylated lysine sites were identified, accounting for about 500 proteins [54,57,58,59,60]. Most of the identified proteins are metabolic enzymes, and among the most identified gene ontologies (GO) are the TCA cycle, branched-chain amino acid degradation, pyruvate metabolism or the fatty acid β-oxidation. Although the functional consequences of protein succinylation are far from being identified for each succinylated protein, there have been reports of increased enzymatic activity (such as for SDHA or the pyruvate dehydrogenase (PDH) complex [59]) and of reduced enzymatic activity (in the case of the Hydroxyacyl-CoA Dehydrogenase Trifunctional Multienzyme Complex Subunit Alpha (HADHA) [61] or the ETC complex I [62]). Many of the succinylated lysines are also targets for other PTMs such as acetylation [59,60]. This crosstalk between the PTMs could represent another way of regulating their deposition. Interestingly, most of the enzymes for the fatty acid oxidation in both peroxisomes and mitochondria were found to be succinylated but the resulting activity was opposite, activating in the peroxisome and inhibitory in the mitochondria [63], reminiscent of the inhibitory role of succinate on mitochondrial β-oxidation [34].

Even though these metabolic pathways are centered on the mitochondria, all the cellular compartments have been shown to be affected by protein succinylation, from mitochondria, to cytosol and nucleus [54,59]. The functional consequences of protein succinylation highly depend on the context. For histones, lysine succinylation, by changing the positive charge to a negative one, induces a decompaction of the chromatin since the DNA is negatively charged [54]. Hypersuccinylation provoked by mutating the SDH complex was shown to result in the accumulation of succinate that was able to exit mitochondria. The succinylation of histones was thus a result of a defective TCA metabolism and an increase in succinate content, linking once more the metabolic state of cells to their epigenetic landscape.

So far, no clear links have been drawn between lysine succinylation and the pluripotent states. The dramatic impact of these modifications on TCA and ETC proteins, along with the strong effect of histone succinylation, could very well in turn regulate the transition between the developmental states. Indeed, the differentiation of induced pluripotent stem cells (iPSCs) into hepatocyte-like cells was already proposed to be in part driven by a differential succinylation of proteins involved in the carbon metabolic pathways [64]. As detection methods for acylated peptides are only emerging, it would be of major interest to profile the succinylome of developing embryos or various pluripotent states to answer these questions.

### 3.2. Succinate as an Epigenetic Landscape Remodeler

2OGXs are a group of enzymes that, in the presence of oxygen and iron, catalyze the transfer of a methyl group from α-ketoglutarate to different substrates. The products of the reaction, in addition to the methylated substrate, are succinate and CO_2_. Because of the formation of succinate as a byproduct, 2OGXs-catalysed reactions are sensitive to the accumulation of intracellular succinate. Among these enzymes, succinate has been shown to decrease the activity of HDMs and the TET family of DNA demethylases [65,66]. For example, loss of SDH activity, leading to the accumulation of succinate throughout the cell, was shown to increase the level of methylated histone 3 lysine 27 (H3K27), due to a decrease in the activity of the HDM Jmjd3 (Jumonji domain-containing protein D3) [67,68]. Similarly, a reduction in the hydroxylation of the 5-methylcytosine (5mC) due to reduced TET activity was later on shown in a context of SDH deficiency [68]. These results were then also reported in the clinic as a “hypermethylator” phenotype of patient samples with SDH deficiency [69].

Early embryo development is known to be the place of dramatic remodeling of the epigenetic landscape, from histone methylation marks to cytosine methylation (extensively reviewed in [70]). It is thus not surprising that an imbalance in the αKG-to-succinate ratio during early development leads to defects in cellular transitions that seem stage dependent. First, it was shown that a high αKG-to-succinate ratio could promote the maintenance in a naïve state, by maintaining low DNA methylation but also low H3K27me3 levels, both crucial for the naïve identity [24]. Accordingly, an increase in succinate concentration was able to reduce the number of alkaline phosphatase-positive or Nanog^high^ colonies that was associated to an increase in DNA methylation. Previous reports have described the impact of succinate on TET to be mostly linked to an increase of methyl CpG islands in polycomb repressive complex 2 (PRC2)-targeted genes [71]. While PRC2 is dispensable for the maintenance of the pluripotent stage, PCR2-deficient cells present a compromised capacity for further differentiation, highlighting the need for epigenetic rewiring in the differentiation processes [72,73,74]. On the other hand, while high αKG-to-succinate ratio promotes the naïve morphology of mESCs [24], it has been shown to favorize the neuroectodermal differentiation of primed cells, an effect related to the increase in 5mC, H3K27me3 and H3K4me3 levels [25]. This complex image of the roles of succinate in in vitro models of ESC developmental progression is in balance with the development of ex vivo mammal embryos. Indeed, the addition of 0.5 mM of succinate in the culture medium increased significantly the proportion of hamster embryos developing to the blastocyst stage [75], but this difference is not observed in bovine embryos, despite a clear increase in 5mC levels [76]. This highlights a context that is species-dependent and remains to be explored for mouse or human embryos.

Along these lines of evidence for the crucial roles of succinate in early development, a recent study revealed that the production of succinate by the invading trophoblast during implantation is required for the establishment of a proper pregnancy and to avoid recurrent spontaneous abortion in humans [77]. On the other hand, an increased concentration of succinate is a marker for gestational diabetes mellitus, linked to hypervascularization in response to the activation of its receptor, SUCNR1 [78]. This highlights the need for a proper balance of the metabolite in the in vivo context.

The inhibitory action of succinate on dioxygenases (like HDM and TET) by the mass action law also includes the inhibition of the prolyl hydroxylases (PHD) enzymes involved in HIF1α and HIF2α degradation. By inhibiting PHDs, succinate thus participates in the stabilization of the HIFα subunits, with subsequent activation of target gene transcription. This phenomenon is thus called pseudohypoxia [79,80]. As mentioned earlier, HIF1α is crucial for the metabolic switch occurring between naïve and primed ESCs. Indeed, HIF1α triggers the transcription of genes involved in glycolysis and in the OXPHOS inhibition, thus providing a major contribution to the metabolic switch observed in the transition [17,18]. By efficiently switching the metabolism from bivalent in naïve to mostly glycolytic in primed, HIF1α stabilization pushes thus ESCs toward the primed stage [17]. Along with that, HIF1α knock-out naïve ESCs fail to properly transition [18]. On the other hand, hypoxia has been shown to reduce the proportion of 2CLCs suggesting that the continuum of early pluripotency stages might be associated with the finely tuned activity of HIF1α, making the picture more complex than expected.

Interestingly, this metabolic switch is also observed during the reprogramming of somatic cells, with bivalent metabolism, to iPSCs, mostly glycolytic. In a similar manner, HIF1α has been shown to control the metabolic switch and enhance the reprogramming efficiency if overexpressed [81].

## 4. Succinate as a Paracrine Effector

Succinate is also known to have paracrine functions on the cellular microenvironment, notably through the G-protein coupled receptor (GPCR) GPR91, that was deorphanized in 2004, when succinate was identified as its natural ligand [82]. GPR91 was then renamed SUCNR1, succinate being is its exclusive endogenous ligand described up to now, with a EC50 in the range of 28–56 µM (depending on the read-out used to assess [Ca^2+^]_i_) [82]_._ This receptor is expressed in a wide range of tissues, although its expression is particularly high in adipose tissues, liver and kidney [83]. In addition, it has been shown that the level of SUCNR1 can be increased in response to exposure to succinate, at least in hematopoietic cells [84].

Since the discovery of SUCNR1, the biological roles of the succinate-SUCNR1 tandem have been explored in a variety of (patho)-physiological conditions. For instance, succinate has been shown to regulate blood pressure through the release of renin by the kidney [82], to activate stellate cells after liver ischemia [84], to trigger retinal angiogenesis [85] and to exert immunomodulatory properties (reviewed in [84,86,87]).

In the seminal paper of He and colleagues who deorphanized GPR91, the first clues on the signal transduction pathways triggered by this activated receptor were unveiled by looking at the most popular transduction pathways activated by most GPCR. Using HEK293 cells expressing the human GPR91, the authors showed that succinate exposure reduces the cyclic adenosine monophosphate (cAMP) increase provoked by forskolin (a synthetic activator of adenylate cyclase (AC)), in a pertussis toxin (PTX)-sensitive manner. As PTX catalyzes the ADP-ribosylation of G_i/o_ alpha subunits, blocking them in their inactive state, it indicates that SUCNR1 triggers a G_i/o_-dependent signaling pathway. In addition, exposure to succinate also triggers an intracellular increase in calcium concentration, as well as inositol phosphate formation, an effect partly inhibited by PTX. This suggests that SUCNR1 activates phospholipase C (PLC) beta through a PTX-insensitive G_q/11_ pathway and through a PTX-sensitive G_i/o_ pathway [82]. In addition, succinate binding to GPR91 was shown to activate extracellular signal-regulated kinase 1/2 (ERK1/2), first in HEK293 cells expressing GPR91 [82] and then in the retinal ganglion cell line RGC-5 [88]. Still in the HEK293 cells, the activation of the G_i/o_ subunit in response to succinic acid reduced the activity of AC thus leading to a decrease in cAMP. The cAMP response downstream of SUCNR1 seems thus cell- and context-dependent [86].

These major signal transduction pathways activated by GPR91 were later on confirmed and extended in other cell contexts with, however, some cell type specificities (Figure 2). For instance, in Madin–Darby Canine Kidney (MDCK) cells expressing SUCNR1, it was shown that the binding of succinate to GPR91 triggers both [Ca^2+^]_i_ increase and ERK1/2 phosphorylation, in G_q/11_ and G_i/o_ dependent pathways [87]. On the contrary, succinate was shown to activate hepatic stellate cells but without inducing an increase in [Ca^2+^]_i_ [89]. GPR91 activation by succinate leads to AMPc-dependent protein kinase (PKA) activation in rat ventricular cardiomyocytes, resulting in apoptosis [90]. In primary cultures of neonatal cardiomyocytes, it was shown that succinate binding to GPR91 activates PLC-dependent inositol triphosphate (IP3) and Ca^2+^ intracellular increases, provoking activation of Calmodulin Kinase II (CaMKIIδ) and ERK1/2 activation [91]. Vascular endothelial cells of the juxtaglomerular apparatus express SUCNR1 and react to succinate exposure by increased [Ca^2+^]_i_, production of nitric oxide and release of prostaglandin E2 (PGE2) [92]. However, the contribution of GPR91 to NO release might be questioned or might be cell type-specific as succinate increases NO secretion in the urothelial cell culture of both C57BL6 and GPR91 KO mice [93].

Obviously, the activation of these major signaling pathways also affects gene expression. For example, in the retinal ganglion cell line RGC-5 exposed to succinate, ERK1/2 and c-Jun N-terminal kinase (JNK) signaling triggered the induction of cyclooxygenase 2 (COX2) and the release of PGE2 in a GPR91-dependent manner [88]. The link between extracellular succinate and prostaglandin synthesis was confirmed in mouse neuronal stem cells, where the prostaglandin-endoperoxide synthase 2 (*Ptgs2*) was the most succinate-induced upregulated gene, in a GPR91-dependent manner [94]. Regarding the calcium branch of the signaling pathways, calcium-activated CaMKIIδ phosphorylates and inactivates histone deacetylase 5 (HDAC5), thereby modulating the expression of several genes in neonatal cardiomyocytes [91].

Beside modifications of gene expression, the activation of GPR91 by succinate also affects mitochondria, at least in cardiomyocytes. Indeed, the activation of PKCδ downstream to GPR91 promotes the translocation to mitochondria of dynamin-related protein 1 (Drp1), a mitochondrial fission regulator. In addition, GPR91 also activates ERK1/2, leading to phosphorylation of MFF. As a result, succinate provokes the mitochondrial fission, in a PKCδ, ERK1/2 and GPR91-dependent manner [95].

A possible role for GPR91 in the early pluripotency states had not been explored until recently, when we demonstrated that increased succinate concentrations, triggered though heme synthesis or SDH inhibition, have been shown to trigger the 2C-like cell reprogramming of mESCs in a SUCNR1-dependent manner [96]. This reveals that the activation of SUCNR1 and its downstream signals could have broad implications in early pluripotency.

First, since the activity of ERK1/2 in ESCs is tightly regulated, repressed in naïve and activated during the transition to the primed stage [97,98,99], the activation of the receptor by succinate and the subsequent ERK1/2 activation could have drastic implications for the progression along the developmental stages. This is in line with the facilitation of the transition by an increased succinate production by the trophoblasts cells, facilitating the naïve-to-primed transition [77].

Second, the production of NO is known to play a role in the maintenance of pluripotency (reviewed in [100,101]). Indeed, low μM of NO have been shown to protect mESCs from death or differentiation following leukemia inhibitory factor (LIF) withdrawal, through a blocked caspase 3 activation, an upregulation of anti-apoptotic genes such as *Bcl2*, and a downregulation of differentiation genes such as *Brachyury* or *Gata4* (GATA Binding Protein 4) [102]. However, this positive role of NO on stemness maintenance is tightly balanced since an increase in its concentration (low mM) induces a decrease in OCT4 and NANOG abundance, a p53-dependent effect. The same balance has also been shown to be crucial during mouse embryo development [103]. NO, the smallest signaling molecule in the cell, is in turn able to regulate different pathways through the formation of cGMP (cyclic guanosine monophosphate) or the formation of reactive nitrogen species (RNS), for example [104]. The nitric oxide (NO) produced in the cytoplasm is also known to stabilize HIF1α, shown to be important for the metabolic switch happening during the naïve-to-primed transition [17,18]. This effect of NO on HIF1α is probably dose-dependent, as NO has also been reported to induce the S-nitrosylation of the cysteine 533 of the HIF-1α protein, decreasing the action of the oxygen-dependent PHD proteins [105].

Third, the production of PGE2 downstream of SUCNR1 could have direct effects on the pluripotent states as it is known to protect mESCs from apoptosis and to increase cell proliferation through prostaglandin receptor-mediated mitogen-activated protein kinase (MAPK) and Akt (protein kinase B) pathway activation [106,107,108]. Depending on the context, PGE2 has been shown to also promote the maintenance of an undifferentiated naïve state [109] or even to favorize differentiation such as in the BMP4-mediated mesodermal differentiation [110].

Finally, in human ESCs, the acquisition and maintenance of the naïve state, long known in mESC, depends on the modulation of various pathways and so far, no consensus has been found. While most of them rely partly on the inhibition of the MAPK [111,112,113,114,115], others include the inhibition of PKC (α, β, γ and δ isoforms) [114] or the inhibition of HDACs [111], two pathways that are modulated in response to the activation of SUCNR1, either positively or negatively, respectively.

## 5. Conclusions

Often disregarded as an important metabolite in the regulation of cell fate, it is becoming increasingly clear that succinate plays a pivotal role in the regulation of embryo development or cellular potential. Originally, succinate was considered mostly under the αKG-to-succinate ratio lens but it is recently taking the center stage by itself. Indeed, through its regulation of the hypoxic response and HIFα stabilization, its contribution to post-translational modifications, its role in the regulation of the methylated epigenetic landscape or even its action as a paracrine signal, succinate can modulate the processes that are fundamental for the initiation of a proper development. Recent literature shows that in vivo, a controlled balance for succinate is required for the establishment of a healthy pregnancy and is overall favorable for the proper pre-implantation development. Further metabolomic characterizations of the developing embryo could reinforce the importance of succinate and maybe its receptor SUCNR1 in development, especially with the use of state-of-the-art 3D models (reviewed in [116,117]). Along with these, in vitro models of early development and stem cell states transitions also offer insights into the deep impact of the metabolite on growth and differentiation capacities.

## Figures and Tables

**Figure 1 metabolites-12-00651-f001:**
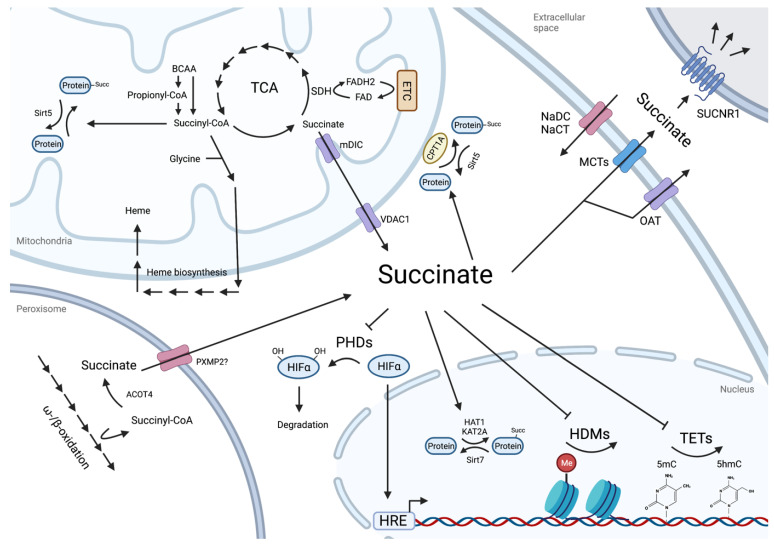
Succinate fluxes and regulatory roles. Figure created with BioRender.com.

**Figure 2 metabolites-12-00651-f002:**
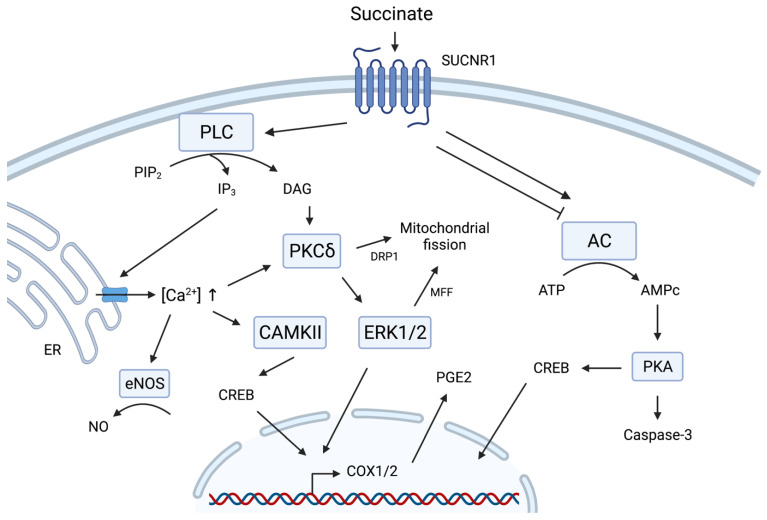
Intracellular signaling triggered by SUCNR1. Depending on the context, SUCNR1 signals lead to either the activation or inhibition of the adenylate cycle (AC), regulating the levels of cyclic adenosyl monophosphate (cAMP). In turn, this metabolite regulates the activity of PKA, leading to the activation of the cAMP response element-binding protein (CREB) and the caspase-3 cell death pathway. SUCNR1 activation also regulates the activity of PLC, able to cleave the Phosphatidylinositol 4,5-bisphosphate (PIP_2_) into diacylglycerol (DAG) and IP_3_ at the plasma membrane. DAG is then capable of activating the protein kinase C (PKC) and subsequent activation of ERK1/2. Together these actors participate in the induction of mitochondrial fission through phosphorylation of the mitochondrial fission factor (MFF) and the recruitment of the dynamin-related protein 1 (DRP1). IP_3_ triggers the release of calcium ions (Ca^2+^) from the endoplasmic reticulum (ER). This Ca^2+^ increase can favorize the activation of PKC, activate the nitric oxide (NO) production by the endothelial nitric oxide synthase (eNOS) and activate the kinase activity CAMKII. This kinase can activate the CREB transcription factor controlling the expression of genes encoding the cyclooxygenases 1 and 2 (COX1/2), among others. These enzymes then produce prostaglandin E2 (PGE2). Figure created with BioRender.com.

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
