# Peer review of "Succinate as a New Actor in Pluripotency and Early Development?"

_metabolites, 2022, doi:10.3390/metabo12070651_

Round 1

Reviewer 1 Report

In this review, the authors enhanced the role of succinate in the early stages of embryonic development and made a very good characterization of the pathways/molecules that succinate modulates. The topic is new and relevant.  

Figures 1 and 2 are clear and resume very well the main pathways involved. 

The references although not very recent were relevant. 

Minor changes: 

- Please add the non-abbreviated form of IMM in line 120 instead of line 143

- Duplicated information: lines 105-108 and 144-146

- Please check the order of references as it seemed to me that what is referenced as 66 corresponds to the 65 in the references section; line 209. 

- lines 275-277: clarify, please.

Author Response

Thank you for reviewing our manuscript.

We have carefully read the manuscript and proceeded to corrections of typo and spelling mistakes.

Regarding your specific comments :

  • Please add the non-abbreviated form of IMM in line 120 instead of line 143

It is now added

  • Duplicated information: lines 105-108 and 144-146

Thank you for having noticed this duplicated information. The information on the probable succinate transporter, PXMP2, was moved from line 147 (first version) to line 188 (new version), and the sentence at line 144-147 (first version) was removed.

We also found a second duplicated information (lines 91-94 and 137-141, initial version). The lines 137-141 have been removed as the contribution of succinyl-CoA to BCAA catabolism and heme biosynthesis was already mentioned in lines 91-94 (first version).

  • Please check the order of references as it seemed to me that what is referenced as 66 corresponds to the 65 in the references section; line 209.

Thank you for this remark. There was indeed a mismatch in the reference list. It is now corrected in this new version of the manuscript.

  • lines 275-277: clarify, please.

The beginning of section 4 (p7) has been reformulated : " Succinate is also known to have paracrine functions on the cellular microenvironment, notably through the G-protein coupled receptor (GPCR) GPR91, that was deorphanized in 2004, when succinate was identified as its natural ligand [83]."

Reviewer 2 Report

The objective of this paper was to provide an up-to-date overview of the role of succinate in early pluripotency and development. 

Some of the in-text citations did not corresponds to the source in the reference list. For example, in Line 164-165, "10 years ago, the succinylation of lysine residues has been shown to occur naturally in vivo 50." It was reference number 49 in the reference list. Also in Line 410 it was mentioned that "(reviewed in 120, 121)" although there were a total of 120 references in the list.  The authors should go through the references cited in the text to make sure that they are consistent with the bibliography. 

Author Response

Thank you for reviewing our manuscript.

We have carefully read the manuscript and proceeded to small corrections of language.

Regarding your specific comment :

  • Some of the in-text citations did not corresponds to the source in the reference list. For example, in Line 164-165, "10 years ago, the succinylation of lysine residues has been shown to occur naturally in vivo 50." It was reference number 49 in the reference list. Also in Line 410 it was mentioned that "(reviewed in 120, 121)" although there were a total of 120 references in the list.  The authors should go through the references cited in the text to make sure that they are consistent with the bibliography. 

Thank you for this important remark. We observed that in the first version of the manuscript, there was a mismatch between the reference numbers in the text and the whole reference list from reference #29 to #120, as reference #29 did not appear in the first version of the text. This has been corrected in this revised version.

Reviewer 3 Report

The review is interesting; however, it needs some changes. On page 4 lines 135 to 144, you are repeating the information that is on page 2 lines 81 to 89.  There are typo mistakes (for example in the Abstract's last line), and a language mistake in the last line of the Abstract.

Author Response

Thank you for reviewing our manuscript.

Regarding your specific comments :

  • The review is interesting; however, it needs some changes. On page 4 lines 135 to 144, you are repeating the information that is on page 2 lines 81 to 89.

The lines 135-144 have been shortened and adapted as the contribution of succinyl-CoA to BCAA catabolism and heme biosynthesis was already mentioned in lines 91-94 (first version). The typo mistakes were corrected and language slightly adapted.

  •  There are typo mistakes (for example in the Abstract's last line), and a language mistake in the last line of the Abstract.

We read carefully the whole manuscript, corrected some spelling errors, and modified the last sentence of the abstract.

Reviewer 4 Report

In this review very important aspect of biochemical activity of succinate is described. Typically, succinate is associated with the tricarboxylic acid cycle, and therefore with a reaction catalyzed by succinate dehydrogenase. Meanwhile, succinate can leave the mitochondrial matrix and function in the cytoplasm as well as in the extracellular space, altering gene expression patterns, modulating the epigenetic landscape, or even displaying hormone-like signaling. This publication seems to be within the scope of journal. However it needs several corrections to be more acceptable for publication.

1.     Please explain the abbreviations used when the term appears for the first time in the text. eg. line 44: OXPHOS; line 61: iPSCs; line 79: NADH; line 120: IMM; line 222: H3K27. Please check carefully whole manuscript.

2.     Please transfer the abbreviation (αKG) from line 73 to line.

3.     Please remove (TCA) from line 135, and “flavin adenine dinucleotide” from line 138 because the FADH2 abbreviation is explained earlier.

4.      In line 138, it should be „(FADH2)”instead of „FADH2”.

5.      In line 159, it should be „carnitine” instead of „Carnitine”.

6.      In lines 162, 339, please remove „Figure created with BioRender.com.”

7.      Line 164: What do the authors mean by saying that “10 years ago, the succinylation of lysine residues has been shown to occur naturally in vivo”? Please correct appropriate part of text.

8.      In lines 279, 308 it should be “Ca2+ instead of “[Ca++]I

9.      In line 348: It should be „histone deacetylase 5” instead of „Histone Deacetylase 5”.

Author Response

Thank you for reviewing our manuscript.

Regarding your specific comments :

  1.     Please explain the abbreviations used when the term appears for the first time in the text. line 44: OXPHOS; line 61: iPSCs; line 79: NADH; line 120: IMM; line 222: H3K27. Please check carefully whole manuscript.

These abbreviations have been explained in their first appearance in the text. We have carefully checked the whole text. Some abbreviations were explained twice (such a MFF, defined in line 355 while it was already defined in line 333 (first version)) or at the second occurence (such as IP3, defined in line while it was already mentioned in line 314). Other abbreviations were missing (such as PSC, in line  48, or JNK, in line 343). All these abbreviations have been corrected or added.

2. Please transfer the abbreviation (αKG) from line 73 to line.

This has been corrected

3. Please remove (TCA) from line 135, and “flavin adenine dinucleotide” from line 138 because the FADH2 abbreviation is explained earlier.

This has been corrected

4. In line 138, it should be „(FADH2)”instead of „FADH2”.

This has been corrected

5. In line 159, it should be „carnitine” instead of „Carnitine”.

This has been modified

6. In lines 162, 339, please remove „Figure created with BioRender.com.”

This has been removed and instead placed rightfully in the figure legends

7. Line 164: What do the authors mean by saying that “10 years ago, the succinylation of lysine residues has been shown to occur naturally in vivo”? Please correct appropriate part of text.

We wanted to stress the point that the natural occurence of succinylation was only demonstrated 10 years ago, while the succinylation of lysine residues after in vitro incubation of proteins with succinic anhydride had been reported 50 years ago (Shiao DD, Lumry R, Rajender S. Modification of protein properties by change in charge. Succinylated chymotrypsinogen. Eur J Biochem. 1972 Sep 18;29(2):377-85. doi: 10.1111/j.1432-1033.1972.tb01999.x. PMID: 5081619).

This is now rephrased in the revised version of the manuscript.

8. In lines 279, 308 it should be “Ca2+” instead of “[Ca++]I

This has been modified

9. In line 348: It should be „histone deacetylase 5” instead of „Histone Deacetylase 5”.

This has been modified